# OpenReview forum: "Energy-Based Adaptive Regularization for Copyright Protection in Language Models"
_ICLR.cc/2026/Conference — Submitted to ICLR 2026_

### Official Review · Reviewer_NZc3 · 2025-10-31

**Soundness:** 3
**Presentation:** 3
**Contribution:** 3
**Rating:** 4
**Confidence:** 4

**Summary:**

The authors frame copyright protection as energy minimization: they observe that unit increments of energy can result in an exponential increase in the likelihood of reproducing copyrighted content. The authors argue the AER procedure balances copyright protection and model utility. The authors produce an interesting body of theory that motivates their methodology and provides theoretical guarantees.

**Strengths:**

S1: The idea of introducing exponential barriers to reduce the chance of regurgitation of copyrighted contents is interesting and novel.

S2: The work focuses on the whole sequence, differently from other anti-memorization procedures that instead focus on single tokens.

S3: The paper is clearly written, and the theoretical results are clearly presented and discussed.

**Weaknesses:**

W1: The experiments are extremely limited in many aspects.

W1.1. The authors only focus on sequence completion on WikiText-2, on windows of 256 tokens. This does not really give an idea of the utility of the produced models on downstream tasks. Could the authors consider some tasks, e.g. like Python inst. MathAbstracts WritingPrompts in [1].

W1.2. While using perplexity is one of the few ways people evaluate the quality of the resulting outputs, it is still a very limited way to assess model utility. Similarly to [1], would it be possible to produce a multi-dimensional evaluation of the quality of the outputs (e.g. Pass@1 for code completion, LLM as a judge for WritingPrompts etc.)?

W1.3: The authors do not compare with almost all the available baselines [1,2]

W2: Despite some claims, the authors do not test against adversarial extraction attempts (e.g. no form of malicious finetuning, poisoning of the training data or simple jailbreaks are performed to evaluate the actual effectiveness of these attack barriers). It is possibl that such barriers may be circumvented through any of these techniques. Could the authors discuss this point thoroughly and provide additional experiments in this regard?


[1] https://arxiv.org/pdf/2412.06619
[2] https://arxiv.org/html/2406.12975v1

**Questions:**

Q1. What happens when windows of larger size are used? How well does the method do both in terms of utility and protection? Could the authors produce experiments in this regard?

Q2: It would be really important to address the limitations of the empirical part of the paper as it would make the work extremely impactful.

I am favourable to increasing the score if the concerns can be addressed.

---

### Official Review · Reviewer_TfXm · 2025-11-01

**Soundness:** 2
**Presentation:** 2
**Contribution:** 3
**Rating:** 2
**Confidence:** 4

**Summary:**

1. The paper tackles the task of copyright protection in large language models, aiming to prevent verbatim reproduction of copyrighted content while keeping model utility. This is challenging because existing methods either remove copyrighted data entirely or use unstable regularization (like inverse loss) that causes gradient explosion and collapse.
2. The proposed Adaptive Energy Regularization (AER) reformulates memorization control as an energy-based optimization problem. The method suppresses verbatim reproduction exponentially with sequence length while maintaining stable gradients.

**Strengths:**

1. The method is supported by strong theoretical analysis, including convergence under the PL condition and exponential suppression guarantees.
2. VRR drops from over 99% to under 1%. The empirical reduction in verbatim copying is striking with only minor perplexity increase.

**Weaknesses:**

1. The writing can be improved. The citation format is awkward (it would be better to use “\citep” in LaTeX). The link between Definition 1 and verbatim copying is unclear. In particular, high energy alone does not guarantee that the model avoids reproducing text segments verbatim.
2. Section 3.2 mentions one related work (referred as the state-of-the-art method), but the discussion of related work is incomplete. There are multiple studies on alternative loss designs, such as [1]. A more comprehensive discussion would strengthen the paper.
3. The paper does not evaluate downstream benchmarks or assess how well the model retains knowledge from copyrighted data. It remains unclear whether the proposed method performs better than simply removing copyrighted material from the pretraining corpus. One way to strengthen the paper is to include evaluations on standard benchmarks such as MMLU and Natural Questions.

[1] Hans et al. (2024) Be like a Goldfish, Don't Memorize! Mitigating Memorization in Generative LLMs

**Questions:**

n/a

---

> ### Comment · Reviewer_dST5 · 2025-11-20
> **A different reviewer's response to this**
>
> I have discussed separate weaknesses in my own review. So this comment is not about my own opinions about whether this paper should or should not be accepted (and should not be read as championing acceptance of this work). However, a rating of 2 seems particularly harsh to me given the level of detail/ contents in this review. Regarding the weaknesses:
>
> 1. A criticism of not using \citep is a nit, not substantive. The link between Definition 1 and verbatim copying could be made clearer, but it also not a deep substantive critique.
>
> 2. This can be addressed with an additional 1-2 sentences in the paper.
>
> Taken together, these are minimal criticisms that read to me like nits. If they form a strong basis for a 2 score, this seems pretty harsh to me, since they are both easily addressable with very minor revisions.
>
> 3. I agree with parts of 3, and have noted specific comments about related work (namely, Cooper et al. (2025), which specifically studies extraction of copyrighted data, and therefore provides a useful test to compare against).
>
> This paper is not about removing copyrighted text from the pre-training corpus; that seems orthogonal in some respects to the questions being studied here. Further, if the aim is to still be able to train on high-quality (copyrighted) data but reduce extraction (as I take is one of the goals here), then removal of such data from the pre-training corpus would arguably not be a relevant point of comparison.

---

### Official Review · Reviewer_dST5 · 2025-11-03

**Soundness:** 2
**Presentation:** 2
**Contribution:** 3
**Rating:** 4
**Confidence:** 4

**Summary:**

The paper proposes Adaptive Energy Regularization (AER), an energy-based framework for mitigating verbatim memorization of copyrighted content. It uses an energy optimization formulation rather than focuses on direct probability suppression.

**Strengths:**

This paper has several strenghts:

- As far as I know, this is novel approach to studying extraction: an energy minimization problem that avoids gradient instability of prior probability-based regularization work. It then is shown to outperform prior work on inverse regularization.

- Shows that the energy gap increases linearly with the regularization weight, confirming the theory. The traiining objective is well-behaved/stable, supported by the theory.

- AER cuts verbatim reproduction from up to 99.1% to <1%, while keeping perplexity low. The results are compelling.

**Weaknesses:**

Please note that I will review any response to this in detail/ am amenable to changing my score if I misunderstood something in my review.

**Problem setup is partially artificial**

The dataset "copyright" subset is synthetic (20% of WikiText-2 labeled as copyrighted). This removes the real-world difficulty of the problem being studied---heterogeneous sources or legal uncertainty, and thereby making the task easier. Why didn't the authors use Llama 1 models, which are known to have been trained on Books3, and study this in practice of a task that better meets the theoretical setup?

The theoretical results depend on clean partitions $\mathcal{C}$ and $\mathcal{O}$, but I don't think those are well-defined in practice.

The assumption that energy can separate copyrighted from ordinary content (See Eq. 6) presupposes what it must prove, so it's unclear to me how the model would "know" which text is copyrighted at scale

**Over-promised "copyright protection"**
[This is easily fixable with more careful writing]

Theorems 7-8 give a notion mathematical suppression of generation probability, but that's not at all the same as legal copyright protection. The model could still produce paraphrases or partial reproductions. That's fine, as a matter of an ML paper making ML contributions. But I think the writing and title risk overclaiming in a domain (copyright) where this is not rigorous or correct. Further, even if it were possible to promise "copyright protection", I don't think the type of controlled experiment done here is sufficiently compelling to support this claim. For carefulness, the paper should phrase results as memorization control, not copyright protection. For more on how hard copyright is re: memorization, see Lee et al. 2023.

**Probabilistic extraction**

Hayes et al. 2025 (probabilistic extraction) and Cooper et al. 2025 (specifically in copyright contexts) show that even when exact recall probability is low, probabilistic reconstruction remains feasible. I'm not sure the paper's exponential suppression argument addresses probabilistic leakage, despite the fact that this type of threat model better aligns with the types of claims that the paper is trying to make about suppression. Without comparison to probabilistic extraction baselines, it’s hard to assess the quality of the intervention super effectively, in my opinion.

**Questions about the theory**

The theoretical definition of "energy-based copyright protection" (Eq. 6-7) assumes a fixed $E_0$ upper bound for "ordinary" data, but I don't think this is enforced or estimated in training? Also I think the exponential guarantee (Theorem 7) assumes the baseline and AER models are trained on the same dataset, yet AER simultaneously treats part of it as "restricted." These conditions seem inconsistent, unless I've misunderstood something.

I think the assumption that the PL condition holds for transformer LLMs is speculative; can you say more about this please?

[1] Lee et al. 2023. Talkin' 'Bout AI Generation: Copyright and the Generative-AI Supply Chain.

[2] Hayes et al. 2025. Measuring memorization in language models via probabilistic extraction

[3] Cooper et al. 2025. Extracting memorized pieces of (copyrighted) books from open-weight language models

**Questions:**

Please see weaknesses above, as I've integrated questions related to observations (which I hope is clearer presented together)

---

### Official Review · Reviewer_LtpV · 2025-11-03

**Soundness:** 2
**Presentation:** 2
**Contribution:** 2
**Rating:** 4
**Confidence:** 4

**Summary:**

The paper proposes an energy-based training objective for copyright compliance in LLMs. The central idea is to interpret the per-token negative log-likelihood as “energy” and to learn a margin between “protected” and “ordinary” text so that the model assigns higher energy (lower probability) to protected sequences. A self-adaptive regularizer is introduced to enforce the desired margin during fine-tuning while attempting to control perplexity degradation. The theory suggests that if a positive energy gap is maintained, the probability of verbatim reproduction decreases exponentially with sequence length. Empirically, the authors report large reductions in a new evaluation metric they call Verbatim Reproduction Rate (VRR) under a simulated setting where 20% of WikiText-2 segments are randomly marked as “protected,” with modest perplexity increases on the remainder. The intended contribution is a training-time mechanism for reducing exact reproduction without heavily hurting utility.

**Strengths:**

The paper offers a clear and mathematically motivated reformulation of “don’t copy protected text” as an energy-margin learning problem. The adaptive regularizer is conceptually simple and likely easy to implement on top of standard fine-tuning pipelines, which is attractive from a practical standpoint. The theoretical analysis, while relying on common smoothness/PL-style conditions, aligns the optimization goal (an energy gap) with a behavioral claim (reduced exact reproduction) and provides an intuitive explanation for stronger suppression on longer sequences. The empirical section is carefully controlled in terms of ablation over the regularization strength and presents results across multiple model families, which helps establish that the effect is not idiosyncratic to a single backbone. The writing is generally accessible and the objective is stated with clarity.

**Weaknesses:**

The empirical methodology does not convincingly demonstrate superiority over established baselines or even comparability to the most relevant literature.

First, the main evaluation uses a simulated “protected” set created by randomly selecting 20% of WikiText-2. This choice does not reflect real copyright risk, which is known to concentrate on rare or highly memorable passages, long-form works, or specific domains. As a result, it is unclear whether the observed gains would transfer to realistic risk distributions or to cross-domain settings.

Second, the paper introduces VRR as the primary metric but does not compare against community-standard overlap measures such as ROUGE-L or LCS, nor does it analyze near-copy/paraphrase leakage, exposure/membership-inference signals, or substantial-similarity style tests. Declaring VRR preferable without head-to-head analyses weakens the empirical claim.

Third, there is no comparison to modern unlearning-based approaches to copyright compliance (e.g., retraining-or-as-if-retrained, distillation to a deletion oracle, selective unlearning with utility preservation, adaptive fusion defenses). Since the proposed method aims to reduce protected-text reproduction while preserving utility, direct comparisons to these lines are essential to establish practical value.

Finally, the related-work coverage appears incomplete: several recent benchmarks and studies on copyright compliance, memorization characterization, and takedown/unlearning are neither cited nor positioned against, making it difficult to place the contribution within the fast-moving landscape.

**Questions:**

I would be substantially more positive if the authors could clarify and, ideally, address the following. How does the method perform when “protected” text is constructed to reflect realistic risk—e.g., long rare passages, book or news snippets, or curated benchmarks—rather than random WikiText-2 segments? Can the authors evaluate VRR alongside LCS/ROUGE-L and a paraphrase-tolerant semantic metric, and report systematic correlations and disagreements among them? Could they compare against representative unlearning baselines under a unified protocol that includes (i) a deletion-oracle reference, (ii) selective unlearning with KL-based utility preservation, and (iii) a recent adaptive-fusion approach, while reporting both leakage metrics and standard utility metrics? What is the behavior under distribution shift, such as evaluating protected material from a domain unseen during the regularizer’s calibration? Finally, can the authors expand the related work to engage with recent compliance benchmarks and studies on memorization and takedown methods, and explicitly state where their objective improves upon or complements those approaches?

Also, there are some relevant works that the authors may need to take a look at.

- Foundation Models and Fair Use
- LLMs and Memorization: On Quality and Specificity of Copyright Compliance
- Copyright Violations and Large Language Models
- SHIELD: Evaluation and Defense Strategies for Copyright Compliance in LLM Text Generation
- Speak, Memory: An Archaeology of Books Known to ChatGPT/GPT-4
- Cpr: Retrieval augmented generation for copyright protection.
- Avoiding Copyright Infringement via Large Language Model Unlearning
- Evaluating Copyright Takedown Methods for Language Models
- Strong copyright protection for language models via adaptive model fusion
- SUV: Scalable Large Language Model Copyright Compliance with Regularized Selective Unlearning
- CopyBench: Measuring Literal and Non-Literal Reproduction of Copyright-Protected Text in Language Model Generation
- A Fictional Q&A Dataset for Studying Memorization and Knowledge Acquisition

---

> ### Comment · Reviewer_dST5 · 2025-11-25
> **Reviewer acknowledgment**
>
> Given that there is no rebuttal/response, I will leave my score and review as-is. I think this paper needs revisions to meet the bar for publication.

---

### Meta-Review · Area_Chair_zK5y · 2026-01-01

**Summary:**

There were many concerns over using a random subset of Wikitext-2 as "copyrighted" text to be protected in the experiments, which is far from realistic and could be problematic. There were also concerns on the metrics used in this paper, as other commonly used metrics like ROUGE-L or LCS are not used, and the utility is only measured using perplexity. Comparison to some important baseline methods are also potentially missing.

**Reviewer Concerns:**

There were no rebuttals posted.

**Reviewer Scores:**

N/A since no rebuttals were posted.

---

### Decision · Program_Chairs · 2026-01-26

Reject